

# Investigating the loads and performance of a model horizontal axis wind turbine under IEC extreme operational conditions

Kamran Shirzadeh [1,2], Horia Hangan [1,3], Curran Crawford [1,4], Pooyan Hashemi Tari [5]

[1] WindEEE Research Institute, University of Western Ontario, London, Ontario, N6M 0E2, Canada
[2] Mechanical and Material Engineering, Western University, London, N6A 3K7, Canada
[3] Civil and Environment Engineering, Western University, London, N6A 3K7, Canada
[4] Mechanical Engineering, Victoria University, Victoria, V8W 2Y2, Canada
[5] Mechanical Engineering, Shahid Beheshti University, Tehran, 19839 69411, Iran

*Correspondence*:  Kamran Shirzadeh (kshirzad@uwo.ca)

**Abstract.** The power performance and loading dynamic responses of a 2.2 m scaled horizontal axis wind turbine (HAWT) under the IEC 61400-1 transient operational extreme conditions were investigated. Extreme wind shears (EWS) and extreme operational gust (EOG) inflow conditions, generated in the WindEEE dome at Western University. The global forces were measured by a multi axis force balance at the HAWT tower base. The unsteady horizontal shear induced a significant yaw
moment on the rotor with similar dynamic loads as the extreme event with no serious effect on the power generation. The EOG severely affected all the performance parameters of the turbine which were highly dependent on the operational TSR and the time duration of the event.

**Keywords:**  Extreme operational Gust; Extreme Shear; HAWT; IEC Standard; Wind Tunnel; Transient Experiment.

## 1.  Introduction

In the past two decades, wind energy has grown to become one of the primary sources of energy being installed worldwide in an effort to reduce greenhouse gas emissions. One of the main factors of this increasing trend is the continued decreasing price of energy generated by wind energy devices. It is still expected for this market to grow by having even lower levelized cost of electricity (LCOE) in the near future (McKinsey & Company, 2019). This price reduction can be facilitated by more
technological advancements (e.g. building larger rotors) and better understanding of the interaction between different wind conditions and the turbines in order to increase the life cycle of these wind energy systems.

The dynamic nature of the atmospheric boundary layers (ABL) affects all the dynamic outputs of the wind energy systems; these all bring challenges to further growth of the wind energy share in the energy sector. One of the main challenges for todays' wind energy systems are the power generation fluctuations which cause instability in the grid network (Anvari et al.,
2016). It has been reported that the effect of the extreme events can get transferred to the grid with even amplification in magnitudes (amount of power generation is related to the cube of wind velocity); the power output of the whole wind farm can change by 50% in just 2 minutes (Milan et al., 2013). These turbulent features also induce fatigue loads on the blades (Burton





et al., 2011) predominantly for the flap wise loadings (Rezaeiha et al., 2017), which then get transferred to the gearbox (Feng et al., 2013), bearings and then the whole structure. Implementation of LIDAR technology can make revolutionary contribution

to this matter by measuring the upstream flow field and give enough time to the control system to properly adjust itself (e.g. blade pitch angles, generator load & etc) in order to reduce overall power fluctuations and the mechanical load variations (Bossanyi et al., 2014).

During the past few decades some comprehensive design guidelines have been developed in terms of load analysis. The International Electrotechnical Commission (IEC) included some deterministic design load cases for commercial horizontal

axis wind turbines (HAWT) in operating condition in the IEC 61400-1 document (IEC, 2005) followed by statistical analysis introduced in the latest edition (IEC, 2019). Herein we test the power generation and loads on a scaled HAWT for representative deterministic gust design conditions as per IEC 2005. In our previous study, (Shirzadeh et al., 2020) the development of the corresponding scaled extreme transient wind fields was carried out. More specifically, the extreme operational gust (EOG) and extreme wind shears (EWS) which includes the extreme vertical and horizontal shears (EVS and

EHS), were all experimentally simulated in the WindEEE dome.

From an aerodynamic perspective the effective angle of attack on the blades and consequently the global lift and drag forces increase during wind gust conditions which result in blade's torque, thrust and root moment amplification. Several experimental studies have been conducted to control the rotor aerodynamics under these transient events. The application of the adaptive camber airfoil in a gusty inflow generated by active grid was investigated by Wester et al., 2018. These type of

airfoils have coupled leading and trailing edge flaps, which can be adjusted to de-camber the profile with increasing lift force. This proved to reduce the integral lift force about 20% at the peak in a gust event. Petrović et al., 2019 developed an algorithm for a PI controller of the pitch angles of a scaled wind turbine in the wind gust conditions generated by an active grid. Using the algorithm, they were able to reduce over speeding of the rotor and the blades' bending moments.

The effect of the wind shears on wind turbine aerodynamics has been studied by several investigators. The effect of various

shear flows and turbulence intensities, generated by active grid, on the near wake region of a small scaled turbine was investigated by Li et al., 2020 using PIV measurements. It has been found that the absolute mean velocity deficit in this region remains symmetric and it is insensitive to the inflow non-uniformity. Also, the mean power production does not change with the amount of the shear. However, the power fluctuation has a linear correlation with the amount of background turbulence intensity, in other words, the effect of higher shears translated as a higher inflow turbulence and therefore higher fluctuations

in power. Similar results were reported by Sezer-Uzol and Uzol, 2013 who used a three-dimensional unsteady vortex-panel method to investigate the effect of transient EWS on the performance of a HAWT. They found that due to the EWS, the blades experience asymmetrical surface pressure variations. Consequently, the rotor produces power and thrust with high amplitude of fluctuations which can cause significant structural issues and reduce the lifetime of the turbine. From the field data perspective it has been reported that for the same reference wind speed, higher turbulence intensities result in relative higher

power efficiencies below the nominal operational condition but the efficiency decreases in transition to rated power (Albers et al., 2007).



Mostly, transient flow fields have been previously generated either numerically or physically by means of active grids. While some of these studies reproduced various transient flows, none had attempted to reproduce and apply the EOG and EWS as per IEC standards and compare the results with stationary and uniform inflow conditions which constitutes the main objectives of the present study. Moreover, the present work employs for the first time a matrix of individually controlled fans to generate customized flow fields and investigate their effect on a model scale wind turbine. The work has been done at the WindEEE dome at Western University Canada. Alongside with the numerical simulations and field data, this setup can contribute to fast development of the new control prototypes of HAWT for customized transient wind effects.

This is an experimental study for examination of the load and power generation of the turbine under four unsteady extreme condition cases (EVS, EHS, negative EVS and EOG), developed in (Shirzadeh et al., 2020). To provide an examination of changes relative to a baseline reference, the results in each case have been normalized with values from a corresponding uniform inflow.

The paper is organized as follows. Section 2 briefly presents the target deterministic operational extreme conditions. Section 3 details the WindEEE chamber and the experiment setups; this section also provides the details about the uniform flow fields used as reference values for comparisons. Section 4 presents the results from EWSs and EOG. Section 5 is dedicated to conclusions.

## 2. Deterministic extreme operating conditions

Prior to introducing the deterministic gust models, it is informative to know how the standard (IEC, 2005) classifies wind turbines based on a reference wind speed and turbulence intensity (TI). The TI in the standard is given for a specific height and is defined as the ratio of the mean standard deviation of wind speed fluctuations to the mean wind speed value at that height, both calculated in 10 min intervals. Three classes of reference wind speeds ($U_{ref}$: I, II and III ) and three classes of turbulence intensities ($I_{ref}$: A, B and C ) are defined that gives a combination of 9 external turbine design conditions that have specified values. One further class for special conditions (e.g. off-shore and tropical storms) is considered which should be specified by the designer. Correspondingly, an extreme wind speed model as a function of height ($Z$) with respect to the hub height ($Z_{hub}$) with recurrence period of 50 years ($U_{e50}$) and 1 year ($U_{e1}$), is defined as follows:

$$U_{e50}(z) = 1.4 U_{ref} \left( \frac{Z}{Z_{hub}} \right)^{0.11},$$
$$U_{e1}(z) = 0.8 U_{e50}(z),$$

(1)

Based on the turbulence class, the streamwise hub height velocity standard deviation ($\sigma_u$), is defined by what is called the normal turbulence model as equation (2).



$$\sigma_u = I_{ref}(0.75\overline{U_{hub}} + 5.6), \tag{2}$$

$\overline{U_{hub}}$ is the average velocity at hub height.

Based on equation (1) and (2) the hub height gust magnitude ($U_{gust}$) is given as:

$$U_{gust} = min\left\{1.35(U_{e1} - U_{hub}); 3.3\left(\frac{\sigma_u}{1+0.1\left(\frac{D}{\Lambda_u}\right)}\right)\right\}, \tag{3}$$

Taking $t = 0$ as the beginning of the gust, the velocity time evolution of the EOG is defined as:

$$U(t) = \begin{cases} \overline{U_{hub}} - 0.37U_{gust} \sin\dfrac{3\pi t}{T}\left(1 - \cos\dfrac{2\pi t}{T}\right); when \quad 0 \le t \le T, \\ \overline{U_{hub}}; \quad when\ t > T\ or\ t < 0, \end{cases} \tag{4}$$

T is the duration of the EOG, specified as 10.5 seconds, $D$ is the diameter of the rotor and $\Lambda_u$ is the longitudinal turbulence scale parameter which is a function of the hub height:

$$\Lambda_u = \begin{cases} 0.7Z_{hub} & for\ Z_{hub} \le 60\ m, \\ 42\ m & for\ Z_{hub} > 60\ m, \end{cases} \tag{5}$$

The EWS can be added to or subtracted from the main uniform or ABL inflows. The EVS velocity time evolution at a specific height ($Z$) can be calculated using equation (6).


$$U_{EVS}(Z,t) = \begin{cases} \left(\dfrac{Z - Z_{hub}}{D}\right)\left(2.5 + 1.28\sigma_u\left(\dfrac{D}{\Lambda_u}\right)^{0.25}\right)\left(1 - \cos\left(\dfrac{2\pi t}{T}\right)\right); when \quad 0 \le t \le T, \\ 0 \quad ; when\ t > T\ or\ t < 0, \end{cases} \tag{6}$$

The EWS duration is 12 s. For a commercial $III_B$ class HAWT with 92 m diameter and 80 m tower hub height, at 10 m/s average velocity, the prescribed EOG and EVS are presented in Figure 1a & b. The time windows in these figures start and end with the extreme event. The standard gust durations in operation condition are relatively long compared to the response

time of scaled wind turbines. Herein, we assume these time durations (10.5 s for EOG and 12 s for EWS) correspond to 4 complete rotor revolutions period in commercial wind turbines which typically have a rotational speed in the range of $\sim$15 − 20 $RPM$; or in another words, the gust time duration is equal to the propagation of the four complete tip vortex loops from a specific blade in the wake. Accordingly, the time scale becomes a function of TSR (i.e. the ratio of the blade tip linear velocity over the free stream), free stream velocity and diameter of the rotor; the length scale is just a function of TSR and diameter of

the rotor. For a scaled wind turbine, the duration corresponding to four revolutions in similar nominal operating conditions is in the order of one second. The experiments in the earlier study showed that the fastest possible gust obtained in the WindEEE dome  with the desired peak factor is around 5 seconds (Shirzadeh et al., 2020). Therefore, it is possible to relevantly decrease the wind speed and TSR to match the parameters up. Assuming a similar $III_B$ class HAWT with hub height of $\sim$2 m with the





2.2 m diameter scaled wind turbine, at 5 m/s average hub-height velocity, the extreme condition profiles look identical to the

full-scale ones (the same peak factor but different gust time) as presented in Figure 1c & d. These are the inflow fields that are

considered in the present experiments. Therefore, based on our assumption the turbine should be working at 1.1 TSR. Also

due to hardware limitation in the physical experiments the EOG has been simplified by excluding the velocity drops before

and after the main peak as it is shown in Figure 1c in red-dashed line.


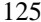

**Figure 1: Extreme operational conditions for a full scale HAWT $III_B$ class with 92m diameter and hub height of 80m at 10 m/s uniform wind speed, (a) extreme operational gust, (b) extreme vertical shear on the rotor with hub height as reference, The scaled extreme condition, (c) the IEC EOG and the simplified EOG that was targeted, (d) extreme vertical shear. Adopted from (Shirzadeh et al., 2020)**





It is worth to mention that these extreme models are relatively simple and not able to capture the real dynamics of the ABL flow-field that directly affect the performance of the turbine (Schottler et al., 2017; Wächter et al., 2012). However, it provides practical guidelines for the development and wind tunnel testing of HAWTs.

## 3. Experimental methodology

### 3.1. WindEEE dome

The experiments were carried out at the Wind Engineering, Energy and Environment (WindEEE) Dome at Western University, Canada. The test chamber has 25 m diameter footprint and 3.8 m height with a total number of 106 fans among which 60 fans mounted along one of the hexagonal walls in a 4×15 matrix and 40 fans are on the rest of the peripheral five
walls (Figure 2a). 6 other larger fans are in a plenum above the test chamber usually being used for 3D flows like tornado and down bursts (Hangan et al., 2017). In the present study, experiments were carried out using the dome in 2D flow (e.g. ABL, uniform straight flows and etc.) closed circuit mode which just the 60-fan wall is operated. In this mode the flow recirculates from the top through the outer shell as it is shown in Figure 2a. Each of these fans are 0.8 m in diameter and are individually controlled to a percentage of their 30-kW maximum power using variable frequency drives. In order to reach higher velocities
at lower fan power set-points at the test chamber for generating EOG a 2D contraction with ratio of 3, was installed just downstream of the 60-fan wall (Figure 2a). These fans are equipped with adjustable inlet guiding vanes (IGV). They can be adjusted stationary from 0% open (close) to 100% open or dynamically in a cycle of opening and closing (Figure 2b). By using this feature the uniform gusts were produced. The transient shears were produced by power modification of the 5 middle fan columns (20 fans). Therefore, contraction walls had no effect on the EWS flow fields.

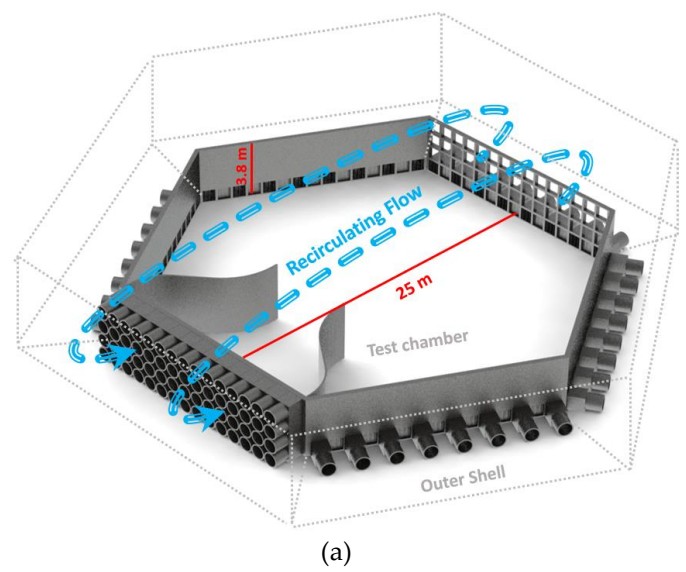

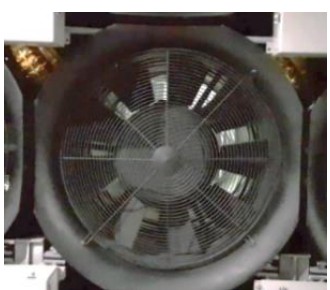

(a)                                                                 (b)



**Figure 2.Simplified geometry of the WindEEE dome, (a) the test chamber and the contraction walls with the flow recirculation path through the outer shell in closed circuit 2D flow mode (b) The adjustable vanes at the inlets of the 60 fans at 70% open vanes**

### 3.2. Experimental setup for power and load performance

To measure the velocity of the flow filed, four cobra probes (TFI Ltd., 2011) are used in a plane 1m upstream (~0.5 D) the rotor and offset 1.3m from the centerline of the primary flow direction. The probes are set at 3m and 0.8m heights corresponding to the highest and lowest heights of the rotating blades' tips (Figure 3). With this configuration the cobra probes can give a proper perception of the flow field over the wind turbine rotor plane. The wind turbine was mounted on a six-
component force balance sensor for measuring all 3 forces/ shears and 3 moments at the base of the tower. After mounting the turbine, the force balance was calibrated as zero. In addition, a light photoelectric diffuse reflection proximity sensor was used which gives a voltage pulse once it detects a light reflection from the blade passing in front. Using the pulse, one can measure the angular velocity of the rotor with a high resolution (three times a revolution). This wind turbine is equipped with a three phase AC generator. A specific electrical circuit was used to convert the voltage and current to DC and feed the power to the
resistors. The last parameter that was monitored was the voltage from the terminals of the power resistors which were set at constant 8.1Ω in order to keep the rotor at the desired TSR (= 1.1). At the end, eight analogue voltage signals (six voltages from force balance, one from proximity sensor and one from load terminals), plus four wind velocity signals gathered to one deck and logged at 2000 Hz frequency for 90s for each experiment run.


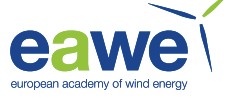
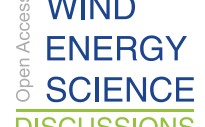

**Figure 3. Setup for measuring power performance and dynamic loads for different types of inflow**

The schematic of the positioning of the wind turbine and the cobra probes relative to the local coordinate system (centre of the WindEEE dome/ base of the tower) is depicted in Figure 4. The shear forces and moments at the base are correlated with the performance of the wind turbine. Based on this local coordinate system, the most important shear force at the base is in the X direction which represents the thrust of the rotor plus the drag force of the tower. The X moment represents torque on the generator plus induced vortex shedding moment; the Y moment shows the bending moment due to drag on the whole structure





(correlated to the forces in X direction). The moment around the Z axis shows the torsion due to horizontal non-uniformity of the flow. The Z force represents the lift on the structure.

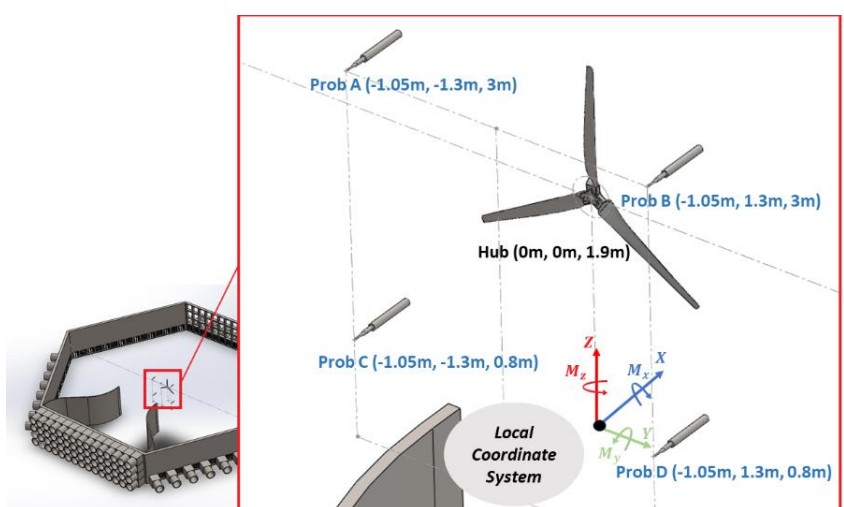

**Figure 4. The arrangement of cobra probes and HAWT relative to local coordinate system**

### 3.3. Baseline uniform inflows

As mentioned earlier, in this study four unsteady extreme condition cases (EVS, EHS, negative EVS and EOG) are considered for the investigation. For the EWSs just the 20 middle fans and in the EOG all of the fans were operated. Some of the results from power performance and loadings of each case are normalized with their corresponding averaged data from one of the two different 5 m/s steady uniform wind inflow (cases A & B in Table 1). The reason for using different uniform cases is due to the difference in the flow characteristics (TI and spectra) related to the different fan setups for each case. At this low TSR, considerable parts of the blades are in stall and the TI magnitude can affect the flow behavior on the suction side of the blades and result in noticeable difference in loads and power performance of the wind turbine. The mean and RMS values of the data obtained from the force balance, turbine power and TSR from these two uniform cases are tabulated in Table 1. The bolded values in this table will be used to normalize the corresponding data from the transient experiment cases. Case A & B for normalizing the EWSs and the EOG respectively.






**Table 1. The mean values of loads and power generation in different steady uniform cases (with the same load, 8.1Ω)**

| 5 m/s Uniform Case | Fan Configuration | Average wind velocity and TI from all the probes | Turbine Performance (mean value \| standard deviation) | | |
|---|---|---|---|---|---|
| A | *5 columns at middle at 40% and IGVs at 100% open* 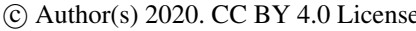 | 4.91 m/s, 16% | | *x-axis* | *y-axis* | *z-axis* |
| | | | Shears (N) **14.36** \| 0.45 | -0.22 \| 0.37 | -4.8\| 0.69 |
| | | | Moments (Nm) **0.87** \| 0.19 | **-20.51** \| 0.91 | -0.14 \| .25 |
| | | | Electrical power (W) | **0.81**\| 0.04 | |
| | | | TSR | 1.30 \| 0.11 | |
| B | *All fans at 30% and IGVs at 10% open* | 5.15 m/s, 10% | | *x-axis* | *y-axis* | *z-axis* |
| | | | Shears (N) **14.39** \| 0.44 | -0.48 \| 0.32 | -5.36 \| 0.71 |
| | | | Moments (Nm) **0.84** \| 0.16 | **-23.06** \| 0.92 | -0.25\| 0.30 |
| | | | Electrical power (W) | **0.67** \| 0.02 | |
| | | | TSR | 1.08 \| 0.08 | |

The turbulence spectra of these two uniform cases are presented in Figure 5. There is a consistent noise from the fans with its harmonics at some specific high frequencies. Due to the steadiness of the flow, large share of the energy is distributed at

the low-end frequencies (i.e. frequencies lower than 3). In this region, the two cases show the same energy distribution. However, for frequencies higher or equal to 3Hz (based on the frozen turbulence hypothesis corresponds to length scales of 1.65m and smaller $\{\frac{1}{3}[s] \times 5 \left[\frac{m}{s}\right] = 1.65\,[m]\}$), difference in energy distribution is noticeable with lower turbulence energy in case B in all the corresponding frequencies relative to case A. All the spectra follow the -5/3 slope consistent with the Kolmogorov theory in inertial subrange (Pope, 2000).






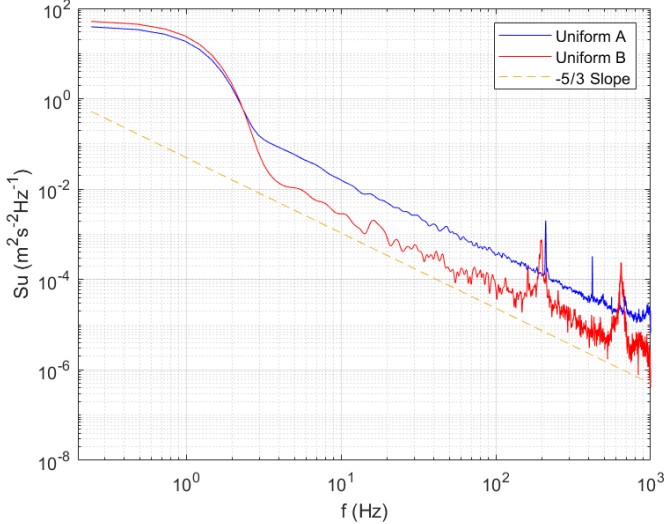

**Figure 5. Comparison of turbulent velocity spectra for the 5 m/s uniform flow cases**

### 3.4. Uncertainty analysis

The epistemic uncertainty of the cobra probes depends on turbulence levels ,but is generally within ±0.5 m/s on average,
up to about 30% turbulence intensity according to the manufacturer company (TFI Ltd., 2011). Considering the 5 m/s average
wind velocity in the experiments the uncertainty of the probes is 10%. The JR3 multi-axis force/torque sensor (75E20S1-
6000N) at the base of the tower has ±0.25% nominal accuracy for all axes. The uncertainly related to measuring and converting
the analog voltages to digital for each signal is negligible.

All the values from the measuring instruments presented in section 4 have been filtered by moving average method except
for the rotor speed. The averaging window for wind velocities, generator voltage and load signals were chosen as 0.2, 0.2 and
0.5 second respectively, which preserve the main shape of signal by just filtering low powered high frequency fluctuations.
Therefore, the RMS values for aleatoric uncertainty averaged in all the experiments in reading the cobra probes and power are
±0.48 m/s and ±0.04 W; the corresponding values for forces and moments in X, Y and Z axis are ±1.13 N, ±0.69 N, ±0.77 N
±1.52 Nm, ±3.17 Nm, ±0.31 Nm respectively. The combined uncertainties in percentage of their corresponding mean values
are tabulated in Table 2. The large percentage error in some of the quantities are due to their very small corresponding average
values.





**Table 2. The combined uncertainty estimation of the measured values averaged in all the experiments**

|  | Epistemic uncertainty | Aleatoric uncertainty | Combined uncertainty |
|---|---|---|---|
| Wind velocity | ±10% | ±9.60% | ±13.86% |
| Power | ±0 % | ±5.71% | ±5.71% |
| X-force | ±0.25% | ±7.86% | ±7.86% |
| Y-force | ±0.25% | ±114.12% | ±114.12% |
| Z-force | ±0.25% | ±15.42% | ±15.42% |
| X-moment | ±0.25% | ±178.21% | ±178.21% |
| Y-moment | ±0.25% | ±13.78% | ±13.78% |
| Z-moment | ±0.25% | ±125.54% | ±125.54% |

## 4.    Turbine test case results

### 4.1. Unsteady EWS

The time history of the results from EVS, negative EVS and EHS cases generated by changing the fan power set-points are presented in Figure 6a , b and c respectively. There are five windows in all of these sub-figures; the first window at the top
shows the filtered wind velocity time histories from the four probes; next window shows the electrical power performance along with TSR; next three windows show the filtered forces and moments time histories exerted at the tower base. The stared axis indexes are normalized by their corresponding values from uniform case A. The second window in all the Figure 6a, b & c illustrates these transient shear cases do not have a significant effect on the overall power performance of the wind turbine. The initial increase and decrease in power productions are just due to the time lag ($\sim 1.5 s$) between the high and the low peaks
of the shears hitting the rotor which is noticeable from cobra probes time history in Figure 6c.

The EVSs (Figure 6a & b) do not have any significant effect on the loads at the base of the tower either. These extreme shears could induce severe fatigue loads at the blade's bearing, blade's root, and the yaw bearing. Having load cells at the blades' roots and nacelle-tower junction or yaw bearing could have given more information about the loads' dynamics and the out-of-plane moment in these scenarios.

The correlation between force in X axis and moment around Y axis is clearly visible in all these figures. The Y force shows random dynamics around zero with no correlation with the forces and moments. The Z force maintained the same value in these EWSs but shows no correlation with other loads. The X moment which represents both the vortex shedding and the torque on the generator has the strongest fluctuation compare to the Y and Z moments time histories. The Z moment in the two EVS cases is a small fluctuating value close to zero due to slight horizontal non-homogeneity of the flow-field.
Theoretically, it should remain the same even at the time EVSs happen but as Figure 6a & b demonstrate there are small variation in the Z moments which can be due to different efficiency of the fans in acceleration and deceleration.

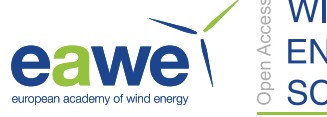

(a) EVS  (b) negative EVS

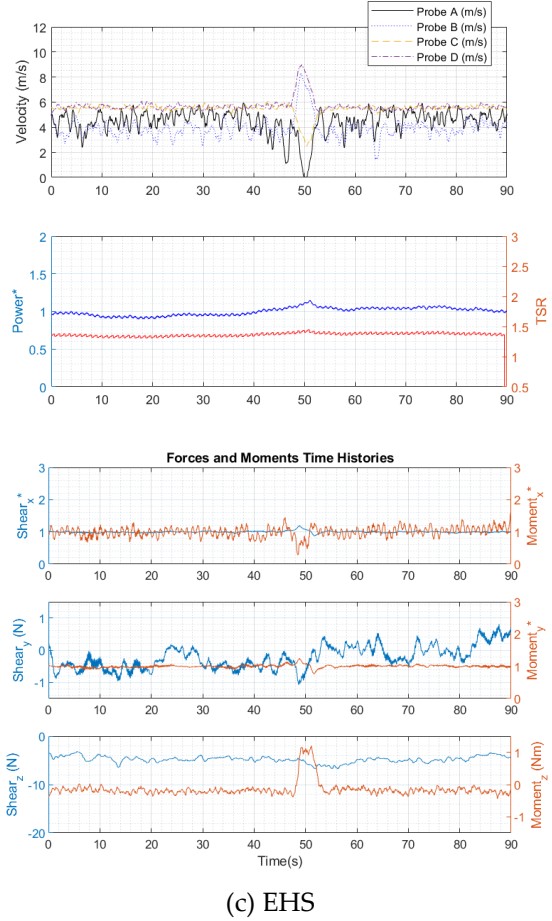

(c) EHS

**Figure 6. The time history of the results from all the measuring instruments in (a) the EVS, (b) the negative EVS and (c) the EHS. The five sub-figures in order from top to bottom in each figure show the wind velocities from 4 cobra probes, the power performance of the turbine, the X, the Y and the Z forces and moments at the base of the tower. The stared axis indexes are normalized by their corresponding value from uniform case A**

In the EHS case, the most important load component at the tower base can be the Z axis moment. The data shows this extreme condition induces a 1.2 Nm torsion on the structure; normalizing this value using equation (7) yields, 0.009414. For 245 a full-scale wind turbine with 92m diameter working in an average 10 m/s wind speed the induced yaw moment on the structure by an EHS event would be 351kNm.

$$CM_z = \frac{M_z}{\frac{1}{2}\rho U^2 AD} \,.$$
(7)

$\rho$ is the density of the air, $A$ is the swept area of the rotor and $D$ is the diameter.



### 4.2. Unsteady EOG

The EOG was generated using IGVs. As the time histories of the measuring instruments suggest this uniform gust has the
most significant effect on both power generation and loads. As the second window in Figure 7 shows, this event can dangerously increase the rotor rotational speed if no active or passive controlling systems are being used (the TSR is changing from ∼1 $to$ 1.33). The electrical power increased 148% at the end of the gust event. The electrical power generation might not be the proper quantity for comparison at these low rotational speeds. The generator efficiency is highly dependent on the rotor speed. Therefore, some part of this significant increase is due to the increase in generator efficiency. The mechanical power
should be a better quantity for comparison, which is highly dependent on the operational TSR. The rotor will generate different amount of torque in different TSRs. Therefore, a similar gust on a similar wind turbine can have different effects in different operating TSRs. The same applies for the loads.

The overall drag on the structure (X shear) and the main bending moment (Y moment) at the base, increased by 105% and 167 % respectively. Their difference depends on the hub height and the rotor diameter; in the current setup the average Y
moment to the average X shear ratio is ∼1.55.

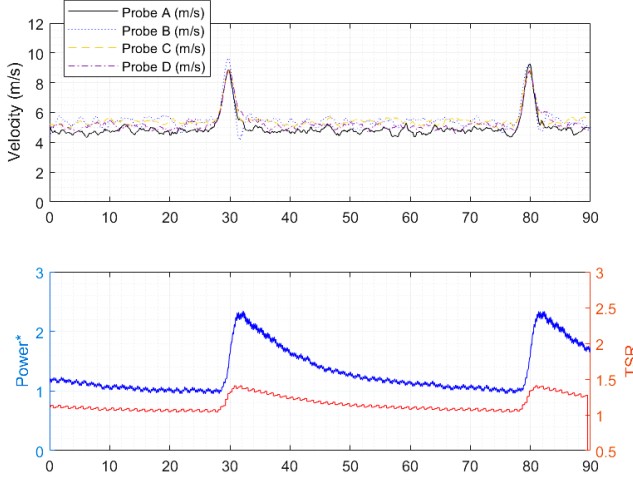



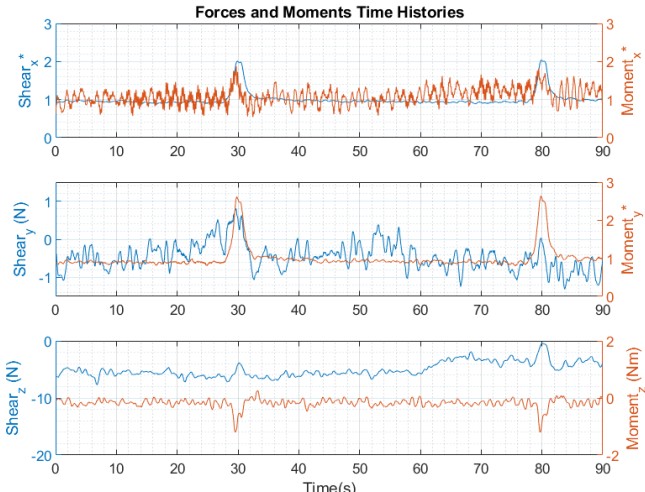

**Figure 7. The time history of the results from all the probes in EOG using IGVs from top to bottom in order, the wind velocities from 4 cobra probes, power performance of the turbine, the forces and moments at the tower base, The stared axis indexes are normalized by their corresponding value from uniform case B**

The loads usually have the same profile as the gust with the same order of rising and falling time. Although, the power generation peak happens at the end of the gust event and then slowly decays afterwards. The extractable energy in a gust with

a specific amplitude and time duration partially accumulates in the rotor rotational momentum with the residue in form of higher instantaneous electrical power generation. After the gust event the stored momentum slowly decays by transforming into electrical power as it can be seen in the second window of the Figure 7.

The gust affects the X moment contributed by the rotor torque. Therefore, the sudden increase in the wind speed (gust) causes abrupt increase in the X moment which has approximately the same rising and falling time as the gust. the Y shear/

force again has no correlation with the gust. There is a slight correlation between Z force and the gust which results in a lift on the turbine (signal moves from negative values toward zero in the last sub-figure in Figure 7). Also, there is a rather large negative correlation between the Z moment and the gust. As the gust happens it pushes and tilts the rotor up which induces a gyroscope moment (~1.2 Nm) on the structure. Note that this highly depends on the structure of the wind turbine and might not happen in commercial HAWTs.

**5.    Conclusions**

An experimental study has been carried out to investigate the effect of transient extreme operating conditions based on the IEC standard (specifically EWSs and EOG) tailored and scaled for a 2.2 m HAWT at the WindEEE dome at Western University. The main assumption used for the length and time scaling is that the duration of each extreme condition is equal to the propagation time of the four tip vortex loops in the wake. Other parameters were adjusted accordingly to accommodate

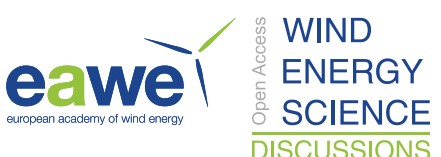

for hardware limitations in generating the flow fields. Two uniform cases as the baselines for comparing the effect of different scenarios were also carried out.

The unsteady EVSs and EHS did not have any significant effect on the power performance and overall loading at the base of the turbine. Nevertheless, EHS induced a noticeable torsion on the structure.

The EOG affects the turbine significantly. Results showed that if no means of control for the rotor speed is considered the
power generation and loadings can increase significantly with a high dynamic behaviour. Also, the reaction of the same wind turbine to the same EOG event can be different depending on the operational TSR. In the EOG event, the loading profiles are corelated with the shape of the gust event itself (the peak of the loads are at the same point as the gust peak), but the power generation's peak happens at the end of the gust event.

Overall, this study presents an alternative experimental procedure for investigating the global loading and power generation
of a scaled wind turbine under scaled deterministic transient wind conditions. The procedure has the potential to be improved and used for developing and testing new wind energy prototypes in transient conditions.

In future work, for the EVSs and EHS cases it is advisable to investigate the loads on the blades' roots and bearings as well as yaw bearing to implement a fatigue load analysis. In the present study the TSR was determined from wake effect scaling and physical limits of the test apparatus, resulting in a TSR~1.1. In future work, an attempt should be made to test at higher
TSR through test apparatus and controller modifications. Testing of the effect of different extreme events durations at different operating TSRs would help validate the suggested time scaling.

**Authors contributions**

KS carried out all the experiments with supervision of HH. KS wrote the main body of the paper with input from all authors.

**Competing interests**

The authors declare that they have no conflict of interest

**Acknowledgements**

All authors thank Gerald Dafoe and Tristan Cormier for helping with the measurement setups. The present work is supported by the WindEEE dome CFI Grant and by NSERC Discovery Grant R2811A03.

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
