# Peer review of "Investigating the loads and performance of a model horizontal axis wind turbine under reproducible IEC extreme operational conditions"

_Wind Energy Science, 2020_

## Referee Comment (RC1) · Rodrigo Soto (Referee) · 17 Dec 2020

Overall, is an interesting paper with high potential. Nevertheless, there are several issues throughout it that needs to be elaborated on before the publication. The main issues to consider are the complete description of the inflow and its reproducibility, and render the conclusions to the results on the paper. I did line to line comments below; I hope these are helpful towards a more complete version. Best regards.

Abstract:

L11: add the corresponding parameter to the length, radius or diameter.

[Figure]

L11-13: It seems that the two sentences were together before, the second one does not sustain itself.

L16: the TSR was not studying to state that. Moreover, is this not always the case?

1. - Introduction

L20-30: note that the word energy is used nine times during these two paragraphs.

L32: the number "2" should be written with letters according to the standard of the Journal, this type of corrections go throughout all the paper (equation -> Eq, figure -> Fig, etc.), you can take a look in the submission settings (https://www.wind-energy-science.net/submission.html

L41: First person is used only a couple of times during the text (we L41, our L42, our L121), like the rest of the manuscript does not, this should be changed.

L47: please check the use of blade's and blades' throughout the manuscript.

L72: This statement is regarding this setup? L42 says that there is a previous work.

L76-79: This is a sentence for the methodology.

2. - Deterministic EOC

L93: please number both equations.

L113: This is unclear. TSR is by definition a function of freestream, rotational speed and radius, so the length scale is only a function of TSR, using the same argument. Is it possible to show this similarity approach by additional equations?

L120: The inflow was just represented by the theoretical profiles and four probes position? This is critical for the study, the gusts were performed several times to study reproducibility?

L122: What are the consequences of this?

L126: ..that these extreme "operational condition" models ... . However, they provide. . .

L129: are there more realistic approaches?

3. - Experimental methodology

L131: The homogeneity of the flow is missing, were the measurements (normal and events) done more than once?

L141: please provide the specific downstream location.

L142: Please consider the word opening instead of open to avoid the open-close confusion.

L146: change filed by field.

L147: What does mean 1.3m from the centerline of the primary flow direction?

L148: It is the case? How are the probes in relation to the fan positioning? Was the inflow studied with more probes?

L151: Please provide details from all the sensors.

L154: specifications are needed. The signal synchronization details are missing. The calibration procedures of the sensors are missing.

L162-168: How is this correlation done?

L163: the most important in terms of? (Also in L243)

L168: Here is stated that the inflow has heterogeneity, how much?

L173: Why only part of the results are normalized? Are then the results comparable?

Table 1: is this TI calculated as the EIC description? AS this shows only the average of the four probes, how scattered are the results between probes? The axis letters are small and Figure 4 shows them in capital. As the mean values are different, the

normalization is done by different values?

L193: this is from only one probe?

L194: which frequencies?

L195: add dimension to the value.

L195-197: this is unclear.

L204: check the comma position.

L207: Is this 0.25% over the full scale range? This needs clarification to neglect it.

L209: There are more than moving average method, which one was used? Please provide a reference.

L211: This is unclear

L215: Please elaborate on this.

Table 2: Is the power epistemic uncertainty 0%? This table should include the values aforementioned in [N] and [Nm]. Clarification in what is the reference to the %. How was the combined uncertainty calculated?

4. - Turbine test case results Clarification of the correlation and its use over the results is needed.

L221: A brief introduction on how the results are presented would improve the understanding of the following sections.

L225: normalized electrical power

L226: starred.

L227: Due to the normalization, as both quantities have uncertainties there is an error propagation, is this considered?

L232-234: This is an outlook that could go at the end of the document.

Figure 6: This can be improved by separating them into three Figures with a)...e) subfigures. As there is no much information or analysis on the steady parts, it would be beneficial zoom in on the event. On the caption: starred. It is possible to perform a frequency spectrum analysis and decouple some influences?

L246: is there some reference that supports the magnitude of this value?

L249: The first sentence should go in the methodology. The second sentence needs to be rephrased.

L252: Is this the case or could be a time delay?

L251-258: From here, it can be inferred that any electrical power data do not provide valuable information? Which is the rated condition of the turbine? Do the conclusions could change with a reduced, rated or overrated operational condition?

L259: How can be stated that these differences depend on the hub height and diameter if they were fixed?

L263: please add a reference.

L264-267: Is this a hypothesis? Is this not contradictory with the electrical power statement on L251-258?

L268: First sentence needs to be rewritten.

L274: Why they might not?

5. - Conclusions

L277: add the corresponding parameter to the length, radius or diameter.

L284: No control was done in this study to state that.

L286: No variation of the TSR was done to conclude this.

---

## Referee Comment (RC2) · Anonymous Referee #2 · 19 Dec 2020

The paper deals with a very interesting topic, moreover with an experimental approach. Some information is needed to clarify the experiments and the relation of these experiments with respect to a full scale turbine. Please find some comments below.

Section 3: No information about the turbine is provided. Could you provide relevant data of turbine design, blade control (collective - IPC), turbine control (fixed speed or controlled speed), operation of the turbine (is it working as a full scale turbine?) and similarity wth full scale. Is it reasonable to make these extreme loads tests with a TSR of 1.1? Is there any similarity with a full scale turbine power extraction? or blade relative wind kinematics? If hte blade is already all stalled there may not be a great sensitivity

[Figure]

on the rotor performance.

Line 147 - typo: filed –> field

Line 156 - The total number of signals should be 12 from Cobras (u,v,w), 6 from force balance,the proximity and the load terminals.

Line 156 - how are you computing TSR? what is your reference velocity?

Figure 3: very low quality image: could you please increase it.

Figure 6 - it could be useful to mark on each subfigure the beginning and the end of the extreme event.

Line 263 - The power generation peak and decay depends on the turbine rotor control, therefore it may be useful to have more details.

---

## Author Comment (AC1) · 7 Jan 2021

Please find a point by point answers to your comments, and the marked-up changes in the manuscript, in the supplement file, thank you.

Please also note the supplement to this comment:
https://wes.copernicus.org/preprints/wes-2020-110/wes-2020-110-AC1-supplement.pdf

---

## Editor Comment (EC1) · Alessandro Bianchini (Editor) · 8 Jan 2021

Dear authors, based on your replies to reviewers' comments, I am encouraging you to submit the revised version of your paper to WES at your earliest convenience. Best regards,

Alessandro

---

## Author Response (AR1)

We thank the referees for their review, comments and suggestions. We have found these comments to be very useful and we have implemented all of them in the revised manuscript and/ or addressed them here in direct reply section. We feel that the paper has been significantly improved by making these changes. The text also has received polishing several times. The major changes in this new revision are: (I) In the result section, we have magnified a 10 seconds window when the events happen to present a better detail about the dynamics of the turbine responses. (II) A reproducibility section has been added which investigate the similarity of the wind velocity profiles in this study compare to the previous study (**Shirzadeh, K., Hangan, H. and Crawford, C.: Experimental and numerical simulation of extreme operational conditions for horizontal axis wind turbines based on the IEC standard, Wind Energy Science, 5(4), 1755–1770, doi:10.5194/wes-5-1755-2020, 2020**.) that shows the current setup reproduces the same events.

We address below each of your comments. At the end of this file the marked-up manuscript with highlighted changes is attached. The location of the corresponding changes is mentioned in each answer to comment.

**RC1**

**Abstract**

- L11: add the corresponding parameter to the length, radius or diameter.

  Ans: diameter has been added.

- L11-13: It seems that the two sentences were together before, the second one does not sustain itself.

  Ans: they have been rearranged

- L16: the TSR was not studying to state that. Moreover, is this not always the case?

  Ans: Correct, that sentence has been excluded.

**1.Introduction**

- L20-30: note that the word energy is used nine times during these two paragraphs.
  Ans: different set of words have been used, thank you.
  - L32: the number "2" should be written with letters according to the standard of the Journal, this type of corrections go throughout all the paper (equation -> Eq, figure -> Fig, etc.), you can take a look in the submission settings (https://www.wind-energyscience. net/submission.html
  Ans: corrections have been considered,
  - L41: First person is used only a couple of times during the text (we L41, our L42, our L121), like the rest of the manuscript does not, this should be changed.
  Ans: corrections have been made

- L47: please check the use of blade's and blades' throughout the manuscript.
  Ans: corrections have been made
- L72: This statement is regarding this setup? L42 says that there is a previous work.
  Ans: That sentence has been deleted. The same setup has been used in this study. new set of words have been used.
- L76-79: This is a sentence for the methodology.
  Ans: to prevent redundancy it has been removed.

**2. Deterministic EOC**

- L93: please number both equations.
  Ans: it has been added.
  - L113: This is unclear. TSR is by definition a function of freestream, rotational speed and radius, so the length scale is only a function of TSR, using the same argument. Is it possible to show this similarity approach by additional equations?
  Ans: The scaling is function of (I) free stream velocity, (II) rotor rotational speed or TSR, (III) rotor size. The corresponding equation has been added. (equation 8)
  - L120: The inflow was just represented by the theoretical profiles and four probes position? This is critical for the study, the gusts were performed several times to study reproducibility?
  Ans: Here the inflow is just presented with the four probes. But in the previous study seven probes in a vertical or horizontal array, depending on the extreme event, were used. Their locations were based on the turbine rotor dimension, two out of the seven probes (probes B & H) were at similar height or lateral distance to the probes in this study. Based on this, a reproducibility analysis has been added just for the flow field. In previous study reproducibility analysis have not been performed, because it included a long process of trial and error for flow modulation and data processing. Therefore, in this study a complementary reproducibility analysis has been added in section 4.3
- L122: What are the consequences of this?
  Ans: more description has been added in line 135. "This simplification stretches the actual rising and falling time, yet, this is the compromise that was made due to hardware limitations. For a wind turbine that operates in a specific average wind, it is the velocity excursion above the average wind speed that is important to capture. More detailed information about the scaling method and the EOCs flow fields are accessible in (Shirzadeh et al., 2020)."
- L126: ..that these extreme "operational condition" models . However, they Provide
  Ans: changes have been made, line 142, Thank you.
- L129: are there more realistic approaches?
  Ans: by simple we mean symmetric and totally deterministic. As far as we know these are the guidelines that are being used by designers and manufacturers. Even though the new version of the standard (2019) uses statistical approach for prescribing the peak factors the generic shapes are still the same.

**3.Experimental methodology**

  - L131: The homogeneity of the flow is missing, were the measurements (normal and events) done more than once?
  Ans: A transitioning sentence has been added. Line 146. The experiments were done once for each event. "As mentioned earlier, in this study similar inflow fields with the ones developed in (Shirzadeh et al., 2020) were reproduced to investigate

the responses of the wind turbine to these scaled transient conditions. For each extreme event only one measurement run was performed. The reproducibility of these events was ensured by direct comparison with the previous study."

- L141: please provide the specific downstream location.

Ans: more details have been added. Line 160

- L142: Please consider the word opening instead of open to avoid the open-close confusion.

Ans: this has been considered, thank you.

- L146: change filed by field.

Ans: it has been fixed.

- L147: What does mean 1.3m from the centerline of the primary flow direction?

Ans: The lateral distance from the center or offset from the center. Different wording has been used. Line 169.

- L148: It is the case? How are the probes in relation to the fan positioning? Was the inflow studied with more probes?

Ans: The flow has been studied in more details in the previous study. For this study we used four probes around the rotor just to see when exactly the event happens in relation with the loads and power generation. This setup also help us to picture the 2D flow field over the rotor without adding more complexity to our experiment setup.

- L151: Please provide details from all the sensors.

Ans: sensor models have been added in the text about proximity sensor and the force balance. More details on Cobra probes is available in the previous study as mentioned in text, Line 174.

- L154: specifications are needed. The signal synchronization details are missing. The calibration procedures of the sensors are missing.

Ans: it all happened in the data acquisition software. Line 185. Also picture of the interface box has been added in figure 3. "analogue voltage signals cables (six voltages from force balance, one from proximity sensor and one from load terminals), plus four wind velocity signals cobra probe cables gathered to one deck, synchronized and logged at 2000 Hz frequency for 90s for each experiment run. All the signals were calibrated as zero in the Turbulent Flow data acquisition software when the 60 fans were off."

- L162-168: How is this correlation done?

Ans: By 'correlated' we meant related. Different wording has been used. line 189

- L163: the most important in terms of? (Also in L243)

Ans: line 191, in terms of magnitude and correlation with performance of the turbine.

- L168: Here is stated that the inflow has heterogeneity, how much?

Ans: this sentence states the Z moment is representing the twist in the structure, which is important in EHS event. However, when fans are uniformly operating there is a slight non-uniformity that induces 0.1 Nm steady yaw moment. Relative to the moment that the EHS induces (1.5 Nm) this is negligible.

L173: Why only part of the results are normalized? Are then the results comparable?

Ans: other parameters were close to zero. Using them to normalize the parameter would have give unreasonably large normal values.

- Table 1: is this TI calculated as the EIC description? AS this shows only the average of the four probes, how scattered are the results between probes? The axis letters are small and Figure 4 shows them in capital. As the mean values are different, the normalization is done by different values?

Ans: TI here are smaller compared to the IEC standard. Matching turbulence was not the focus of these studies. When using all the fans with contraction walls the flow characteristics are uniform. However, in creating EWS when we just used the 20 fans in the middle the flow close to the ceiling is more turbulent. More details can be found in the previous study. The axes letters have been switched to capital.

Yes, We used Case A to normalize EWSs (EVS, negative EVS, EHS) and case B to normalize EOG.

- L193: this is from only one probe?

Ans: line 223, it represents the spatial averaged (i.e. averaged from all the probes)

- L194: which frequencies?

Ans: frequencies have been added, line 225.

- L195: add dimension to the value.

Ans: dimension added.

- L195-197: this is unclear.

Ans: line 227, different wording was used.

- L204: check the comma position.

Ans: done, thanks.

- L207: Is this 0.25% over the full scale range? This needs clarification to neglect it. "The JR3 multi-axis force/ torque sensor (75E20S175E20S4-6000N) at the base of the tower has ±0.25% nominal accuracy of the measuring value for all axes".

Ans: JR3 force balances are very sensitive. Clarifications have been added. Line 239.

- L209: There are more than moving average method, which one was used? Please provide a reference.

Ans: reference has been added. Line 250.

- L211: This is unclear

Ans: based on the criteria described in (Chowdhury, J., Chowdhury, J., Parvu, D., Karami, M. and Hangan, H.: Wind flow characteristics of a model downburst, in American Society of Mechanical Engineers, Fluids Engineering Division (Publication) FEDSM, vol. 1, American Society of Mechanical Engineers (ASME)., 2018.) the moving average time window was chosen to just filter low energy fluctuations from the main signal, so if you plot these fluctuating part of the signal separately the mean value of them is zero.

- L215: Please elaborate on this.

Ans: line 254, they are essentially zero with some fluctuations.

- Table 2: Is the power epistemic uncertainty 0%? This table should include the values aforementioned in [N] and [Nm]. Clarification in what is the reference to the %. How was the combined uncertainty calculated?

Ans: the voltage signal from generator was directly connected to the interface box so essentially the only uncertainty is analog to digital conversion. Mean values of the parameters have been added to the table, clarifications have been added , line 252. Combined uncertainty calculation in line 251.

**4. Results**

- L221: A brief introduction on how the results are presented would improve the understanding of the following sections.

Ans: now the result subfigures have window number, introduction has been provided, line 262- 266.

- L225: normalized electrical power

Ans: changed, thank you.

- L226: starred.

Ans: changed, thank you.

- L227: Due to the normalization, as both quantities have uncertainties there is an error propagation, is this considered?

Ans: This have been neglected as now mentioned in line 254.
- L232-234: is an outlook that could go at the end of the document.

Ans: It has been moved to line 363.
- Figure 6: This can be improved by separating them into three Figures with a): : :e) subfigures. As there is no much information or analysis on the steady parts, it would be beneficial zoom in on the event. On the caption: starred. It is possible to perform a frequency spectrum analysis and decouple some influences?

Ans: four times magnified windows have been added to the figures, thank you. We have done frequency analysis for highly fluctuating X-moment to see if there are any peaks related to vortex shedding so we can filter that. There were some energy peaks but we couldn't find any relation between those frequency with vortex shedding and rotor rotation frequency. As the focus of this study is to measure the global loading and power generation we chose not to include any frequency analysis here .
- L246: is there some reference that supports the magnitude of this value?

Ans: there is a very interesting CFD simulation of a 87 m diameter turbine (Cai, X., Gu, R., Pan, P. and Zhu, J.: Unsteady aerodynamics simulation of a full-scale horizontal axis wind turbine using CFD methodology, Energy Conversion and Management, 112, 146–156, doi:https://doi.org/10.1016/j.enconman.2015.12.084, 2016.). The yaw moment in the typical yaw condition as they calculated is about 30 to 90 Nm as blades rotate. Comparisons have been added in line 300.

- L249: The first sentence should go in the methodology. The second sentence needs to be rephrased.

Ans: this has been done, thank you.
- L252: Is this the case or could be a time delay?

Ans: there is no delay in EOGs, the flow is perfectly uniform.
- L251-258: From here, it can be inferred that any electrical power data do not provide valuable information? Which is the rated condition of the turbine? Do the conclusions could change with a reduced, rated or overrated operational condition?

Ans: The nominal TSR for this turbine is 5. It is known that generators in low rotor speeds have very high hysteresis losses which can bring their efficiency as low as 20-30 %. Therefore it would have been better if we were able to directly measure the mechanical power. As it has been mentioned in the future works, more investigation in various operational TSRs needs to be performed to asses the scaling method results with what actually happens in full scale. The conclusions definitely can change with various operational TSR and actually that is one of the main points of the proposed scaling method to relate this result to a full range of full scale conditions.
- L259: How can be stated that these differences depend on the hub height and diameter if they were fixed?

Ans: We meant that Y-moment is the moment of X-force with the lever-arm close to the hub height. The sentence was not necessary and it has been deleted.
- L263: please add a reference.

Ans: this is based on our results. clarifications have been added in line 321.

- L264-267: Is this a hypothesis? Is this not contradictory with the electrical power statement on L251-258?

Ans: This is the analysis of the results. Essentially turbines in various TSR have various capabilities in extracting energy from the gust. Some part of the energy in the gust get stored in form of angular momentum with excess instantaneously transformed in power generation. After the gust, that stored momentum energy gradually transforms to power generation. As it can be seen figure 7 window II.

- L268: First sentence needs to be rewritten.

Ans: different wording has been used. Line 227.

- L274: Why they might not?

Ans: Because they may have different/stiffer structure.

**5. Conclusions**

- L277: add the corresponding parameter to the length, radius or diameter.

Ans: done, thank you.

- L284: No control was done in this study to state that.

Ans: correct, different wordings have been used. Line 366.

- L286: No variation of the TSR was done to conclude this.

Ans: correct, different wordings have been used. Line 368.

**RC2**

**Section 3:**

- No information about the turbine is provided. Could you provide relevant data of turbine design, blade control (collective - IPC), turbine control (fixed speed or controlled speed), operation of the turbine (is it working as a full scale turbine?) and similarity wth full scale. Is it reasonable to make these extreme loads tests with a TSR of 1.1? Is there any similarity with a full scale turbine power extraction? or blade relative wind kinematics? If hte blade is already all stalled there may not be a great sensitivity on the rotor performance.

Ans: this turbine is the exact turbine that has been examined in (Refan, M. and Hangan, H.: Aerodynamic Performance of a Small Horizontal Axis Wind Turbine, Journal of Solar Energy Engineering, 134(2), doi:10.1115/1.4005751, 2012.) it does not have any active control. "The turbine has 1 kW rated power at 12 m/s wind speed and nominal TSR of 5" line 182.

We are using a new scaling method which has been elaborated in the previous study. Supplementary explanation has been added (equation 8). The main similarity considered, is the wake kinematics compared to full scale. The blades are probably stalled in this low TSR, and you are correct, our results are dependent on the TSR. As mentioned in future works the previously proposed scaling method needs further investigation.

- Line 147 - typo: filed –> field

Ans: changed, thank you.

- Line 156 - The total number of signals should be 12 from Cobras (u,v,w), 6 from force balance,the proximity and the load terminals.

Ans: there is a misunderstanding here, figure 3 has been re arranged to better present the setup. From each cobra probe there is one cable and it is actually the software that process the data and gives (u,v,w).

- Line 156 - how are you computing TSR? what is your reference velocity?

Ans: the averaged velocity from all the probes is the reference velocity. Clarifications have been added in Line 182. Thank you.

- Figure 3: very low quality image: could you please increase it.

Ans: it has been re arranged with new details.

- Figure 6 - it could be useful to mark on each subfigure the beginning and the end of the extreme event.

Ans: it has been re arranged, thank you.

- Line 263 - The power generation peak and decay depends on the turbine rotor control, therefore it may be useful to have more details.

Ans: as it has been mentioned in line 308, no means of control has been used. In this case based on our results the power peak happens at the end of the EOG, and its decay purely depends on rotor inertia and the amount of stored angular momentum at the end of the EOG.

[revised manuscript text omitted]

---

## Referee Report (RR1)

Investigating the loads and performance of a model horizontal axis wind turbine under reproducible IEC extreme operational conditions

- The force balance information is misleading, a clarification of this is necessary.
- For completeness the information about the turbine must be included such as, blades airfoil-shape, twisted? Tapered?
- For a smooth correction, please check that the number of the line that is referred agree with the document, it was really difficult to follow the author's answer, probably it was written and then change something which mismatched all the lines.
- Technical issues, in the writing, were not addressed, such as equation-> Eq. figures -> Fig, etc.

**JR3 Multi-Axis Force-Torque Sensor Technical Specifications**

| Sensor Model:
Mechanical Load Rating: | 75E20S4
650 lb | 75E20S4
1300 lb |
|---|---|---|
| Diameter (in) | 7.50 | 7.50 |
| Thickness (in) | 2.00 | 2.00 |
| Material | 15-5PH SS | 15-5PH SS |
| Weight (lb) | 15.0 | 15.0 |
| Nominal Accuracy, all axes (% measuring range) | ±0.25 | ±0.25 |
| Operating Temp. Range, non-condensing (°F) | -40 to +150 | -40 to +150 |
| **$F_x$, $F_y$** | | |
| Standard Measurement Range (lb) | ±650 | ±1300 |
| Digital Resolution (lb) | 0.081 | 0.16 |
| Stiffness (lb/in) | 0.98e6 | 1.6e6 |
| Single-axis Overload (lb) | 4150 | 7600 |
| Multi-axis Overload Coefficient, a (lb) | 4450 | 7850 |
| Multi-axis Overload Coefficient, b (lb) | 4150 | 7600 |
| **$F_z$** | | |
| Standard Measurement Range (lb) | ±1300 | ±2600 |
| Digital Resolution (lb) | 0.16 | 0.32 |
| Stiffness (lb/in) | 7.61e6 | 12.0e6 |
| Single-axis Overload (lb) | 12,500 | 24,100 |
| Multi-axis Overload Coefficient, c (lb) | 12,500 | 24,100 |
| **$M_x$, $M_y$** | | |
| Standard Measurement Range (in-lb) | ±5000 | ±9800 |
| Digital Resolution (in-lb) | 0.63 | 1.23 |
| Stiffness (in-lb/rad) | 38.2e6 | 64.4e6 |
| Single-axis Overload (in-lb) | 19,900 | 39,500 |
| Multi-axis Overload Coefficient, d (in-lb) | 19,900 | 39,500 |
| **$M_z$** | | |
| Standard Measurement Range (in-lb) | ±5000 | ±9800 |
| Digital Resolution (in-lb) | 0.63 | 1.23 |
| Stiffness (in-lb/rad) | 11.9e6 | 21.6e6 |
| Single-axis Overload (in-lb) | 17,000 | 32,300 |
| Multi-axis Overload Coefficient, e (in-lb) | 17,000 | 32,300 |

-
-

---

## Author Response (AR2)

Please find point by point response to your comments in the following:

5 • **The force balance information is misleading, a clarification of this is necessary.**

We have contacted the JR3 company about the accuracy of their sensor and we recognized that we are dealing with high uncertainties in our values. As per our request, they made a calibration test just for the Y moment (Fig. 1). They exerted plus and minus 58.75 Nm on it two times each. The result show +-0.12 Nm accuracy in average which is a lot lower than what the

10 datasheet suggested. As the company previously mentioned to us those values (datasheet) are the worst possible accuracy that you might expect. However, many changes have been made to recognize this in the text including removing all the smaller loads with unacceptable uncertainties. Now just the X force and Y moment are being presented. For horizontal shear, the additional Z moment has been presented to show the yaw moment on the structure mentioning this value associates with a large uncertainty. The turbine itself is heavy and relatively large so installing it on a smaller and more sensitive force balance

15 was not possible.

The changes made regarding this comment can be found marked up version: lines 211-224, the additional table 2 includes all the uncertainty based on the datasheet. Line 233-239. Table 3 now just presents the X force and Y moment. The Y moment uncertainty has been calculated based on the calibration results. lines 264-270 have been removed which was related to loads in the other axes. Caption in Fig. 6 has been modified. In Fig.6c an additional window presenting Z moment has been added

20 while recognizing the probable high uncertainty. Lines 294-300 have been removed which were related to the other forces. Caption in the Fig. 7 has been modified.

[Figure]

**Fig. 1: Email from JR3 company with the calibration on the Y moment**

- **For completeness the information about the turbine must be included such as, blades airfoil-shape, twisted? Tapered?**

A detailed study on this exact turbine has been performed by (Refan, M. and Hangan, H.: Aerodynamic Performance of a Small Horizontal Axis Wind Turbine, Journal of Solar Energy Engineering, 134(2), doi:10.1115/1.4005751, 2012.) which includes the blade geometry and power curve of the turbine. This has been added to the text. In line 161 in the marked up version.

- **For a smooth correction, please check that the number of the line that is referred agree with the document, it was really difficult to follow the author's answer, probably it was written and then change something which mismatched all the lines.**

Sometimes converting the document to PDF readjusts the lines. This time everything has been double checked in PDF format.

- **Technical issues, in the writing, were not addressed, such as equation-> Eq. figures -> Fig, etc.**

All the fonts and figures and equations captions have been edited. You can find this changes in marked up version : lines 88, 90, 99, 101, 111, 116, 119, Fig.1, 138, 141, 144, 146, Fig. 2, 155, Fig.3, 172, Fig. 4, Fig. 5, 247, 273, 275, Fig.6, 291, Fig. 7, .